# Spatial control of neuronal metabolism through glucose-mediated mitochondrial transport regulation

Anamika Agrawal[1], Gulcin Pekkurnaz[2]*, Elena F Koslover[1]*

[1]Department of Physics, University of California, San Diego, San Diego, United States; [2]Section of Neurobiology, Division of Biological Sciences, University of California, San Diego, San Diego, United States

**Abstract** Eukaryotic cells modulate their metabolism by organizing metabolic components in response to varying nutrient availability and energy demands. In rat axons, mitochondria respond to glucose levels by halting active transport in high glucose regions. We employ quantitative modeling to explore physical limits on spatial organization of mitochondria and localized metabolic enhancement through regulated stopping of processive motion. We delineate the role of key parameters, including cellular glucose uptake and consumption rates, that are expected to modulate mitochondrial distribution and metabolic response in spatially varying glucose conditions. Our estimates indicate that physiological brain glucose levels fall within the limited range necessary for metabolic enhancement. Hence mitochondrial localization is shown to be a plausible regulatory mechanism for neuronal metabolic flexibility in the presence of spatially heterogeneous glucose, as may occur in long processes of projection neurons. These findings provide a framework for the control of cellular bioenergetics through organelle trafficking.

DOI: https://doi.org/10.7554/eLife.40986.001

*For correspondence:
gpekkurnaz@ucsd.edu (GP);
ekoslover@physics.ucsd.edu (EFK)

**Competing interests:** The authors declare that no competing interests exist.

## Introduction

Cellular metabolism comprises an intricate system of reactions whose fine-tuned control is critical to cell health and function. A number of quantitative studies have focused on metabolic control through modulating reactant and enzyme concentrations and turnover rates (*Grima and Schnell, 2006*; *Amar et al., 2008*). However, these studies generally neglect the spatial organization of metabolic components within the cell. By localizing specific enzymes in regions of high metabolic demand (*Laughton et al., 2007*; *Zecchin et al., 2015*), as well as clustering together consecutively acting enzymes (*O'Connell et al., 2012*), cells have the potential to substantially enhance their metabolism.

Spatial organization is particularly critical in highly extended cells, such as mammalian neurons, whose axons can grow to lengths on the meter scale. Metabolic demand in neurons is spatially and temporally heterogeneous, with especially rapid ATP turnover found in the presynaptic boutons (*Rangaraju et al., 2014*), and ATP requirements peaking during synaptic activity and neuronal firing (*Shulman et al., 2004*; *Ferreira et al., 2011*; *Weisová et al., 2009*). Neurons rely primarily on glucose as the energy source for meeting these metabolic demands (*Peppiatt and Attwell, 2004*). Due to the long lengths of neural processes, the glucose supply can vary substantially over different regions of the cell (*Ferreira et al., 2011*; *Weisová et al., 2009*; *Hall et al., 2012*). In myelinated neurons, for instance, it has been speculated that glucose transport into the cell is localized primarily to narrow regions around the nodes of Ranvier (*Magnani et al., 1996*; *Harris and Attwell, 2012*; *Rosenbluth, 2009*), which can be spaced hundreds of microns apart (*Ibrahim et al., 1995*; *Butt et al., 1998*). Glucose transporters in neurons have also been shown to dynamically mobilize to active synapses, providing a source of intracellular glucose heterogeneity (*Ashrafi et al., 2017*). Furthermore,

**eLife digest** Cells are equipped with power factories called mitochondria that turn nutrients into chemical energy to fuel processes in the cell. Hundreds of mitochondria move throughout the cell, shifting their positions in response to energy demands. This happens via molecular motors that pick the mitochondria up and carry them to new locations. Such movements enable the mitochondria to accumulate in parts of the cell with the greatest energy needs.

Mitochondria of nerve cells or neurons have a particular challenging job, as neurons can be very long and different parts within the cells can have different energy needs. It has been shown that mitochondria stop in regions where nutrients such as sugar are most concentrated. So far, it has been unclear whether this regulated stopping helps control energy balance in neurons.

Here, Agrawal et al. used a computational model of rat neurons to find out whether sugar levels are sufficient in guiding mitochondria. The results showed that the mitochondria only accumulated in high-nutrient regions when the sugar concentrations were moderate – not too low and not too high. A specific range of sugar levels was necessary to make this mechanism useful for increasing the efficiency of energy production. Such concentrations match the ones observed in healthy rat brains.

When neurons are unable to meet their energy demands, they stop working and sometimes even die. This is the case in many diseases, including diabetes, dementia, and Alzheimer's disease. Computer models allow us to explore the complex energy regulation in detail. A better understanding of how neurons regulate their energy production and demand may help us discover how they become faulty in these diseases.

DOI: https://doi.org/10.7554/eLife.40986.002

varying levels of activity in the mammalian brain may lead to varying extracellular glucose levels, resulting in spatially heterogeneous nutrient access (*Hawkins et al., 1979*). Individual axons have been shown to span across multiple regions of the brain (*Matsuda et al., 2009*), enabling them to encounter regions with different glucose concentrations.

Most ATP production in neurons occurs within mitochondria: motile organelles that range from interconnected networks to individual globular structures that extend throughout the cell. As energy powerhouses and metabolic signaling centers of the cell, mitochondria are critical for neuronal health (*Nunnari and Suomalainen, 2012*). Their spatial organization within the neuron plays a pivotal role in growth and cell physiology (*Li et al., 2004*). Defects in mitochondrial transport are involved in the pathologies of several neurological disorders such as peripheral neuropathy and Charcot-Marie-Tooth disease (*Baloh, 2008*; *Baloh et al., 2007*).

A number of studies have shown that mitochondria are localized preferentially to regions of high metabolic demand, such as the synaptic terminals (*Li et al., 2004*; *Chang and Reynolds, 2006*). Such localization can occur via several molecular mechanisms, mediated by the Miro-Milton mitochondrial motor adaptor complex that links mitochondria to the molecular motors responsible for transport (*Mishra and Chan, 2016*). Increased $Ca^{2+}$ levels at active synapses lead to loading of calcium binding sites on Miro, releasing mitochondria from the microtubule and thereby halting transport (*Wang and Schwarz, 2009*; *Macaskill et al., 2009*). High glucose levels can also lead to stalling, through the glycosylation of motor adaptor protein Milton by the glucose-activated enzyme *O*-GlcNAc transferase (OGT) (*Pekkurnaz et al., 2014*). This mechanism has been shown to lead to mitochondrial accumulation at glucose-rich regions in cultured neurons (*Pekkurnaz et al., 2014*). It is postulated to regulate mitochondrial spatial distribution, allowing efficient metabolic response to heterogeneous glucose availability.

Mitochondrial positioning relies on an interplay between heterogeneously distributed diffusive signaling molecules (such as $Ca^{2+}$ and glucose), their consumption through metabolic and other pathways, and their effect on motor transport kinetics. While the biochemical mechanisms and physiological consequences of mitochondrial localization have been a topic of much interest in recent years (*MacAskill and Kittler, 2010*; *Mishra and Chan, 2016*), no quantitative framework for this phenomenon has yet been developed.

In this work we focus on glucose-mediated regulation of mitochondrial transport, developing quantitative models to examine the consequences of this phenomenon for metabolism under

spatially varying glucose conditions. Our approach relies on a reaction-diffusion formalism, which describes the behavior of species subject to both consumption and diffusion. Reaction-diffusion systems have been applied to describe the spatial organization of a broad array of cellular processes (**Kondo and Miura, 2010**), ranging from protein oscillations in *E. coli* (**Howard et al., 2001**), to coordination of mitotic signalling (**Chang and Ferrell, 2013**), to pattern formation in developing embryos (**Bunow et al., 1980**; **Gregor et al., 2005**). The response of actively moving particles to spatially heterogeneous, diffusive regulators has also been extensively investigated in the context of chemotaxis (**Van Haastert and Devreotes, 2004**). In contrast to most chemotactic cells, however, mitochondria have no currently known mechanism for directly sensing glucose gradients. Instead, they are expected to accumulate in response to local glucose concentration only. Our goal is to delineate the regimes in which such a crude form of chemotaxis can lead to substantial spatial organization and enhancement of metabolism.

Specifically, we model the modulation of mitochondrial density with glucose concentration in a tubular axonal region, focusing on two forms of spatial heterogeneity. In one case, we consider an axonal domain between two localized regions of glucose entry, representing the internodal region between nodes of Ranvier in myelinated neurons (**Figure 1a**). The second case focuses on an unmyelinated cellular region with continuous glucose permeability, embedded in an external glucose gradient (**Figure 1b**). In both cases, we show that mitochondrial accumulation and enhanced metabolic flux is expected to occur over a limited range of glucose concentrations, which overlaps with physiological brain glucose levels. Our simplified quantitative model allows identification of a handful of key parameters that govern the extent to which glucose-mediated mitochondrial halting can modulate metabolism. We establish the region of parameter space where this mechanism has a substantial effect, and highlight its potential importance in neuronal metabolic flexibility and ability to respond to spatially varying glucose.

## Results

### Minimal model for mitochondrial and glucose dynamics

We begin by formulating a quantitative model to describe the spatial localization of mitochondria that halt in a glucose-dependent manner, in the presence of localized sources of glucose. This

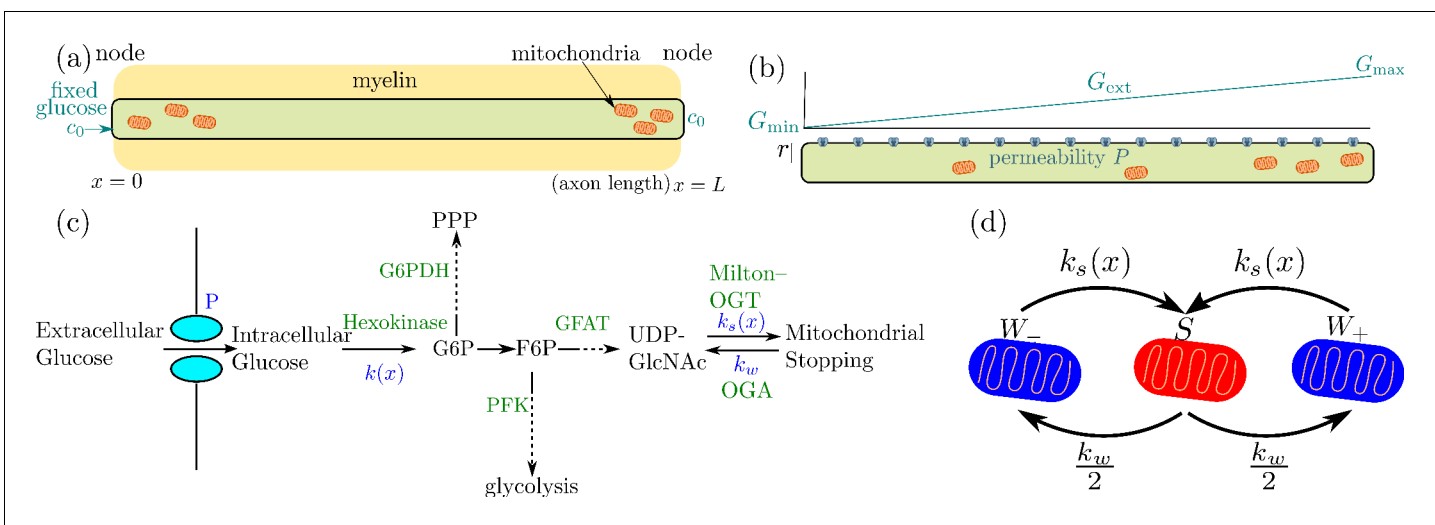

**Figure 1.** Schematic diagram of a simplified model for glucose-mediated mitochondrial transport regulation. (**a**) Myelinated axonal region, with glucose entry localized at the nodes of Ranvier. Mitochondria accumulate at nodes due to the higher glucose concentration (**b**) Unmyelinated axonal region, subject to a linear glucose gradient. Glucose permeability is uniform throughout, with mitochondrial accumulation occuring at the region of high external glucose (**c**) Key steps of the metabolic pathway linking glucose availability and mitochondrial halting. (**d**) Mitochondrial transport states and rates of transition between them ($W_{\pm}$ represents retrograde and anterograde motion, $S$ represents the stationary state).
DOI: https://doi.org/10.7554/eLife.40986.003

situation arises in myelinated neurons, which have glucose transporters enriched at the nodes of Ranvier, leading to highly localized sources of glucose spaced hundreds of micrometers apart within the cell (*Saab et al., 2013*).

Neuronal glucose transporters are known to be bidirectional (*Simpson et al., 2007*), allowing glucose concentration within the cell to equilibrate with external glucose. For simplicity, we assume rapid transport of glucose through these transporters, so that the internal concentration of glucose at the nodes where transporters are present is assumed to be fixed. The cellular region between two glucose sources is modeled as a one-dimensional interval of length $L$ with glucose concentration fixed to a value $c_0$ at the interval boundaries (*Figure 1a*). Glucose diffuses throughout this interval with diffusivity $D$, while being metabolized by hexokinase enzyme in the first step of mammalian glucose utilization (*Figure 1c*) (*Wilson, 2003*).

The concentration of glucose is thus governed by the reaction-diffusion equation,

$$\frac{dG}{dt} = D\frac{\partial^2 G}{\partial x^2} - k(x)G(x) \tag{1}$$

where $k(x)$ describes the spatial distribution of the hexokinase enzyme as well as the rate of consumption. In the case of spatially uniform, linear consumption [$k(x) = k$, a constant] this equation can be solved directly, yielding a distribution of glucose that falls exponentially from each source boundary, with a decay length $\lambda = \sqrt{D/k}$ (*Kholodenko, 2006*).

Hexokinase 1 (HK1), the predominant form of hexokinase expressed in neurons, is known to localize preferentially to mitochondria (*John et al., 2011*), which in mammalian axons can form individual organelles approximately 1 μm in length (*Fawcett, 1981*). We carry out numerical simulations of *Equation 1* where consumption is limited to locations of individual discrete mitochondria, represented by short intervals of length Δ. Specifically, we define the mitochondria density as $M(x) = n(x)/(\pi r^2 \Delta)$, where $n(x)$ is the number of mitochondria overlapping position $x$, and $r$ is the axon radius. The phosphorylation of glucose by mitochondrial hexokinase is assumed to follow Michaelis-Menten kinetics, described by

$$k(x) = \frac{k_g M(x)}{G(x) + K_M} \tag{2}$$

where $K_M$ is the saturation constant and $k_g$ is the turnover rate of glucose (per unit time per mitochondrion). The turnover rate $k_g$ incorporates both the catalytic rate of hexokinase and the number of hexokinase enzymes per mitochondrion. This expression reduces to the case of constant linear consumption when glucose concentration is low ($G \ll K_M$) and mitochondria are uniformly distributed throughout the region.

In general, glucose consumption depends on the location of mitochondria within the domain. Mitochondrial distribution in neurons is known to be mediated through regulation of their motor-driven motility (*Chang and Reynolds, 2006*; *Pekkurnaz et al., 2014*). Individual mitochondria switch between processively moving and paused states, modulated by the interplay between kinesin and dynein motors and the adaptor proteins that link these motors to the mitochondria (*Schwarz, 2013*). In our model, we simulate mitochondria as stochastically switching between a processive walking state that moves in either direction with velocity $v$ and a stationary state. The rate of initiating a walk ($k_w$) is assumed to be constant, while the halting rate ($k_s(x)$) can be spatially heterogeneous. For simplicity, we assume the mitochondria are equally likely to move in the positive (+) or negative (-) direction each time they initiate a processive walk (*Figure 1b*).

It has recently been demonstrated that the key motor adaptor protein (Milton) is sensitive to glucose levels, halting mitochondrial motility when it is modified through O-GlcNAcylation by the OGT enzyme (*Pekkurnaz et al., 2014*). Our model employs a highly simplified description of mitochondrial dynamics, which assumes that all pauses are associated with such an O-GlcNAcylation event. Recovery from the pause at the constant rate $k_w$ corresponds to removal of the modification through the activity of the complementary enzyme O-GlcNAcase (OGA). Although there is evidence indicating long-term glucose deprivation can reduce OGA expression (*Zou et al., 2012*), for simplicity we assume in our model that OGA activity is independent of glucose levels. In vivo axonal mitochondria have been observed to undergo short-lived sporadic pausing while continuing to move processively in their previous anterograde or retrograde direction (*Russo et al., 2009*; *Wang and Schwarz,*

*2009*). Such pauses are subsumed into an effective processive velocity $v$ in our model. Other sources of pausing, such as Ca$^{2+}$-regulated motor disengagement, PINK1/Parkin-mediated detachment of motors, and anchoring to the microtubules by syntaphilin (*Schwarz, 2013*), are not considered here in order to focus specifically on the effect of glucose-dependent mitochondrial spatial organization.

Upon entry into the cell, the first rate-limiting step of glucose metabolism is its conversion into glucose-6-phosphate by hexokinase. Further downstream metabolic pathways split, with much of the flux going to glycolysis while a small fraction is funneled into the pentose phosphate pathway and the hexosamine biosynthetic pathway (HBP). The HBP produces UDP-GlcNAc, the sugar substrate for O-GlcNAcylation (*Figure 1c*) (*Hart et al., 2011*). In our model, we assume that the rate of UDP-GlcNAc production equals the rate of glucose conversion by hexokinase, scaled by the fraction of G6P that is channeled into the hexosamine biosynthetic pathway. This assumption is valid if, at each point of pathway branching, the Michaelis-Menten saturation constants for the two branches are similar. This in fact appears to be the case for both the branching of the pentose phosphate pathway and glycolysis from the hexosamine biosynthetic pathway which is the focus of this work (see Appendix 2). Consequently, saturation of the initial glucose conversion step will imply saturation of the entire hexosamine biosynthetic pathway. We therefore model the kinetics of Milton modification using the same Michaelis-Menten form as for hexokinase activity, with the pathway flux leading to Milton modification subsumed within a rate constant for mitochondrial stopping ($k_s$).

We note that the subcellular organization of the intermediates in the conversion from glucose into O-GlcNAcylated Milton is largely unknown. In our model, we make the extreme case assumption that all intermediates are localized to mitochondria, with only the initial glucose substrate capable of diffusing through the cytoplasm. We note that cytoplasmic diffusion of any of the pathway intermediates would attenuate the effect on mitochondrial localization. Our simplified model thus gives an upper limit on the extent to which mitochondria can localize at high glucose regions through the Milton modification mechanism. Following these simplified assumptions, we treat the kinetics of mitochondrial halting as dependent only on the local glucose concentration, according to the functional form

$$k_s(x) = \frac{k_s G(x)}{G(x) + K_M} \tag{3}$$

where $K_M$ is the Michaelis-Menten constant of hexokinase.

We proceed to evolve the simulation forward in time, with glucose consumption localized to regions within $\pm \Delta/2$ of each discrete mitochondrial position (details in Materials and methods). A snapshot of one simulation run is shown in *Figure 2a*, highlighting the accumulation of stationary mitochondria in the high glucose regions near the ends of the domain.

We are interested primarily in investigating the steady-state distribution of mitochondria and glucose in this system, averaged over all possible mitochondrial trajectories. We thus proceed to coarse-grain our model by treating the distribution of mitochondria as a continuous field $M(x) = W_+(x) + W_-(x) + S(x)$, where $W_+(x)$ is the distribution of mitochondria walking in the positive direction, $W_-(x)$ is the distribution of those walking in the negative direction, and $S(x)$ is the distribution of stationary mitochondria. We can then write down the coupled differential equations governing the behavior of the mitochondrial distributions as:

$$\frac{dW_+}{dt} = -v\frac{\partial W_+}{\partial x} - k_s(x)W_+ + \frac{k_w S}{2}$$

$$\frac{dW_-}{dt} = v\frac{\partial W_-}{\partial x} - k_s(x)W_- + \frac{k_w S}{2} \tag{4}$$

$$\frac{dS}{dt} = k_s(x)[W_+ + W_-] - k_w S.$$

The glucose distribution evolves according to *Equation 1* with consumption rate $k(x)$ given by *Equation 2*. The boundary conditions at the ends of the domain are assumed to be reflective for the mitochondrial distributions, and to have a fixed glucose concentration $c_0$. The stationary state for this system can be calculated numerically (see Materials and methods). The formulation with a continuous mitochondrial density faithfully represents the behavior of simulations with discrete mitochondria, as illustrated in *Figure 2b*.

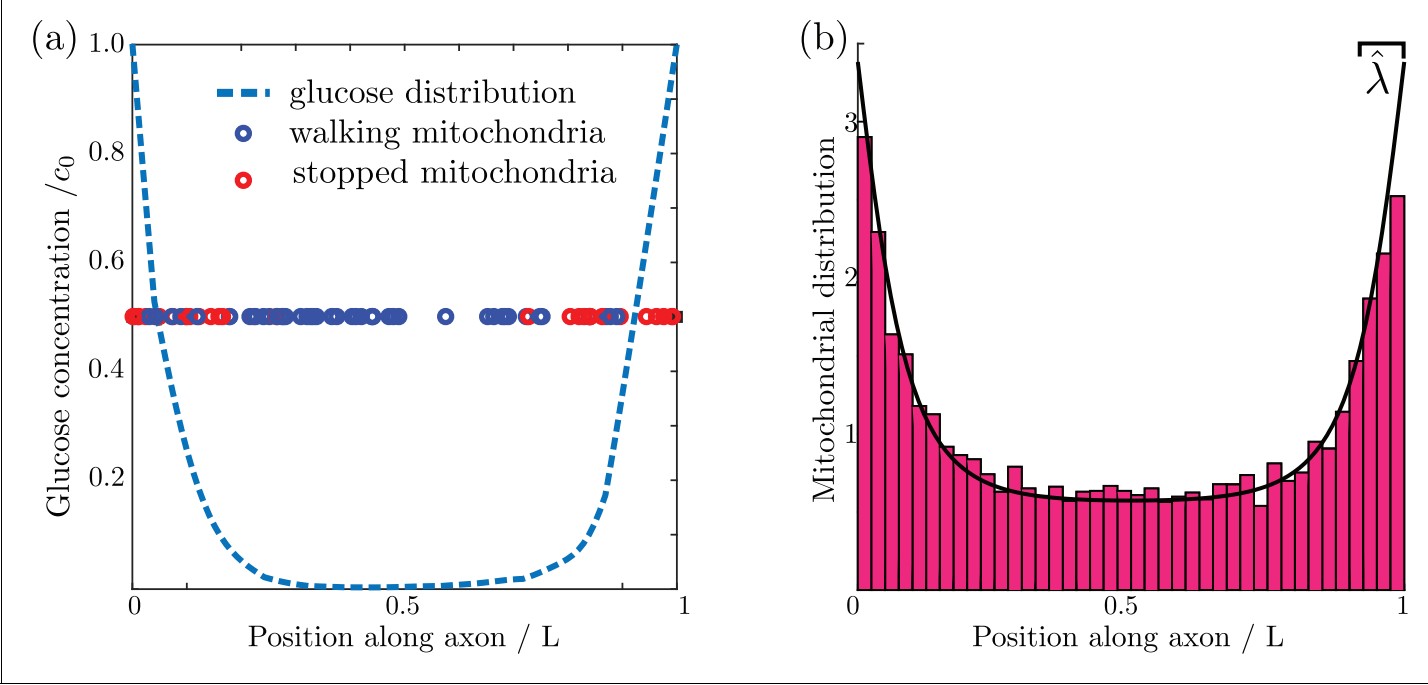

**Figure 2.** Mitochondrial and glucose distributions from simulations with discrete motile mitochondria. (a) Glucose distribution and positions of individual mitochondria (b) Normalized mitochondrial distribution, $M(x)/\overline{M}$, obtained from simulating discrete mitochondrial motion (histogram compiled from 100 independent simulations), compared to numerical calculation of steady state continuous mitochondrial disribution (black curve). Results shown are for parameter values: $\hat{\lambda} = 0.08$, $\hat{c}_0 = 1$, $\hat{k}_s = 100$.

DOI: https://doi.org/10.7554/eLife.40986.004

The steady-state spatial distribution of mitochondria and glucose in the continuous system depend on six parameters: $k_s/k_w, K_M, c_0, D, L, k_g\overline{M}$ where $\overline{M}$ is the average mitochondrial density in the axon (number of mitochondria per unit volume) . Estimates of physiologically relevant values are provided in *Table 1*. Dimensional analysis indicates that three of these parameters can be used to define units of time, length, and glucose concentration, leaving three dimensionless groups. We choose to use the following three dimensionless parameters, each of which has an intuitive physical meaning:

**Table 1.** Physiological parameter values estimated from published data.

| | | |
|---|---|---|
| Cytoplasmic glucose diffusivity | $D$ | 140 μm²/s |
| glucose turnover per mitochondrion | $k_g$ | $1.3 \times 10^5 \text{s}^{-1}$ |
| axon radius | $r$ | 0.4 μm |
| internodal distance | $L$ | 250 μm |
| mitochondrial density | $\overline{M}$ | 0.3 μm⁻³ |
| hexokinase Michaelis-Menten constant | $K_M$ | 0.03 mM |
| brain glucose levels | $c_0$ | 0.7 − 1.3 mM |
| ratio of stopped to moving mitochondria at high glucose | $k_s/k_w$ | 19 |
| glucose permeability | $P$ | 20 nm/s |
| glucose transporter (GLUT3) Michaelis-Menten constant | $K_{MP}$ | 3 mM |

Source: see Appendix 1 for details of parameter estimates.

DOI: https://doi.org/10.7554/eLife.40986.005

$$\hat{\lambda} = \sqrt{\frac{DK_M}{k_g \overline{M} L^2}}, \quad \hat{c}_0 = \frac{c_0}{K_M}, \quad \hat{k}_s = \frac{k_s}{k_w} \tag{5}$$

Here $\hat{\lambda}$ is the length-scale of glucose decay relative to the domain length, $\hat{c}_0$ is the boundary glucose concentration relative to the saturation constant $K_M$, and $\hat{k}_s$ is the ratio of stopped to walking mitochondria at high glucose levels. We proceed to explore the steady-state distribution of mitochondria and glucose as a function of these three parameters.

## Mitochondrial localization requires limited range of external glucose

In order for mitochondria to preferentially accumulate at the source of glucose via a glucose-dependent stopping mechanism, three criteria must be met. First, the glucose concentration needs to be higher at the source than in the bulk of the cell, as occurs when the decay length due to consumption is much smaller than the size of the domain ($\hat{\lambda} \ll 1$). Second, if glucose levels become too high ($\hat{c}_0 \gg 1$) then both glucose consumption rates and stopping rates of the mitochondria become saturated, leading to a flattening of glucose and mitochondrial distributions (*Figure 3*). There is thus an upper limit on the possible external glucose concentrations that will yield mitochondrial localization at the edges of the domain. Finally, the mitochondria must spend a substantial amount of time in the stationary state, since walking mitochondria will be broadly distributed throughout the domain. Because the stopping rate is itself dependent on the glucose concentration, this criterion implies that very low concentrations will also not allow mitochondrial localization. *Figure 3* shows the distribution of glucose and mitochondria at different values of the external glucose $\hat{c}_0$, illustrating that accumulation of mitochondria at the edges requires intermediate glucose levels.

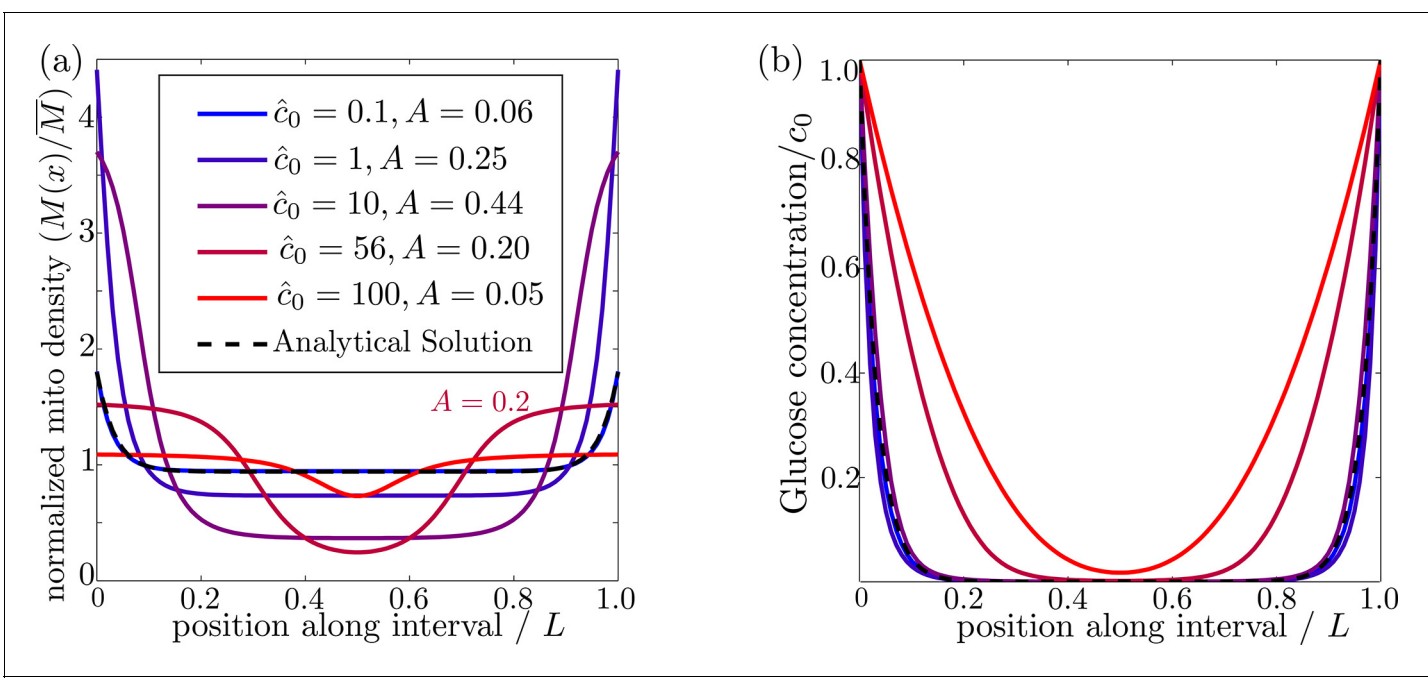

**Figure 3.** Effect of external glucose concentration on intracellular glucose and mitochondrial distributions. (a) Normalized mitochondrial distribution $(M(x)/\overline{M})$, for different values of edge concentration $\hat{c}_0$. The curve with $\hat{c}_0 = 56$ illustrates the accumulation cutoff $A = 0.2$. (b) Glucose distribution normalized by edge concentration $(G(x)/c_0)$. The black dashed line in both panels indicates the analytical solution for the low glucose limit (Materials and methods, *Equation 13*). Source data provided in '*Figure 3—source data 1*'.
DOI: https://doi.org/10.7554/eLife.40986.006

The following source data is available for figure 3:

**Source data 1.** Matlab code to calculate and plot steady-state distributions with localized glucose entry.
DOI: https://doi.org/10.7554/eLife.40986.007

To characterize the distribution of mitochondria along the interval, we introduce an accumulation metric $A$, defined by

$$A = 6\sigma^2/L^2 - 0.5$$

where $\sigma^2$ is the variance in the mitochondrial distribution. This metric scales from $A = 0$ for a uniform distribution to $A = 1$ for two narrow peaks at the domain edges. Mitochondrial distributions with several different values of the accumulation metric are shown in *Figure 3a*. We use a cutoff of $A = 0.2$ to define distributions where the mitochondria are localized at the glucose source.

We explore the dependence of the mitochondrial accumulation on the three dimensionless parameters defining the behavior of the system: the stopping rate constant $\hat{k}_s$, the glucose decay length $\hat{\lambda}$, and the external concentration $\hat{c}_0$. Because only the stopped mitochondria localize near the glucose sources, increasing the fraction of mitochondria in the stopped state (increased $\hat{k}_s$) inevitably raises the overall accumulation (*Figure 4a*). The fraction of mitochondria in the stopped state will depend on both $\hat{k}_s$ and the overall levels of glucose, as dictated by $\hat{c}_0$ (*Figure 4b*). Experimental measurements indicate that at high glucose concentrations, approximately 95% of mitochondria are in the stationary state (*Pekkurnaz et al., 2014*). We are thus interested primarily in the parameter regime of high stopping rates: $\hat{k}_s \geq 10$. The limited range of concentrations that lead to mitochondrial accumulation at the edges of the domain can be seen in *Figure 4a*.

For a high stopping rate ($\hat{k}_s = 10$), we then calculate the mitochondrial accumulation as a function of the remaining two parameters: $\hat{\lambda}, \hat{c}_0$. Here, again, we note that only intermediate glucose concentrations result in accumulation, with the range of concentrations becoming narrower as the decay length $\hat{\lambda}$ becomes comparable to the domain size (*Figure 4c*). We can establish the concentration range within which substantial accumulation is expected, by setting a cutoff $A = 0.2$ on the accumulation metric and calculating the resulting phase diagram (*Figure 4d*). Below the lower concentration cutoff, insufficient mitochondria are in the stationary state and so no localization is seen. This lower cutoff disappears in the limit of infinite $\hat{k}_s$. At intermediate concentrations, mitochondria are localized near the domain edges. Above the upper concentration cutoff, no localization is observed due to saturation of the Michaelis-Menten kinetics.

Using empirically derived approximations for the rate of glucose consumption by mitochondria and the diffusivity of glucose in cytoplasm (see *Table 1*), we estimate the decay length parameter as $\hat{\lambda} \approx 0.03$. The mitochondria are then expected to localize near the glucose source only if $\hat{c}_0 < 66$. Because the saturation concentration for hexokinase is quite low ($K_M \approx 0.03\text{mM}$) (*Wilson, 2003*), we would expect mitochondrial accumulation for glucose concentrations below about 2 mM. We note that physiological brain glucose levels have been measured at $0.7 - 1.3$ mM, depending on the brain region (*McNay et al., 2001*), implying that glucose-dependent halting of mitochondrial transport would be expected to result in localization of mitochondria at nodes of Ranvier.

## Glucose-dependent halting can increase metabolic flux under physiological conditions

Localizing mitochondria to the glucose entry points is expected to increase the flux of glucose entering the cell, thereby potentially enhancing the overall metabolic rate. We calculate the overall effect of transport-based regulation on the net metabolic flux within the simplified model with localized glucose entry. *Figure 5* shows the effect of increasing mitochondrial stopping rates ($\hat{k}_s$) on the total rate of glucose consumption in the interval between nodes of glucose influx. At low $\hat{k}_s$ values, mitochondria are distributed uniformly throughout the interval. At high $\hat{k}_s$ values and at sufficiently low glucose concentrations, the mitochondria cluster in the regions of glucose entry, increasing the overall consumption rate by up to 40% at physiologically relevant glucose levels ($c_0 = 1$ mM). We note that in hypoglycemic conditions, glucose levels can drop to 0.1 mM (*Silver and Erecińska, 1994*), further increasing the magnitude of this effect.

In the case of limited glucose transport into the cell, intracellular glucose levels could be significantly below the concentrations outside the cell. Measurements of intracellular glucose in a variety of cultured mammalian cell types indicate internal concentrations within the range of $0.07 - 1\text{mM}$, up to an order of magnitude lower than glucose concentrations in the medium (*John et al., 2008*).

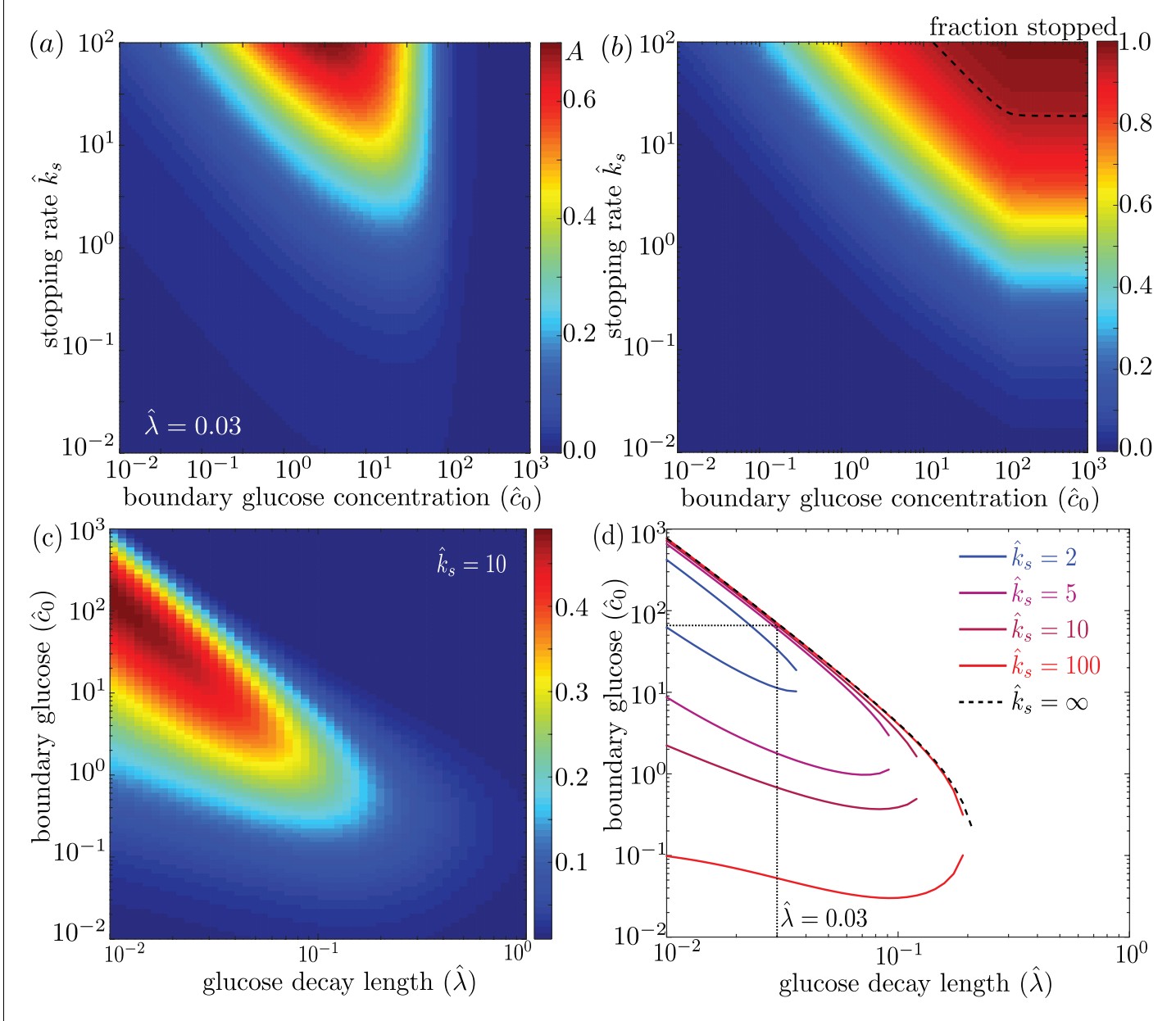

**Figure 4.** Effect of model parameters on mitochondrial accumulation at regions of localized glucose entry. (a) Accumulation metric as a function of boundary glucose levels and mitochondrial stopping rate. (b) Fraction of mitochondria in the stopped state. Black dashed line indicates parameters corresponding to 95% stopped mitochondria. (c) Accumulation metric as a function of glucose levels $\hat{c}_0$ and decay length $\hat{\lambda}$. (d) Phase diagram for mitochondrial accumulation, showing upper and lower concentration cutoffs for accumulation above the cutoff of $A_{\text{cut}} = 0.2$. Dashed black line shows limit of high stopping rate $\hat{k}_s$. Dotted black line indicates estimate of $\hat{\lambda}$ for physiological parameters, and corresponding upper concentration cutoff. Source data provided in '*Figure 4—source datas 1–3*'.

DOI: https://doi.org/10.7554/eLife.40986.008

The following source data is available for figure 4:

**Source data 1.** Matlab code to calculate and plot mitochondrial accumulation with localized glucose entry.
DOI: https://doi.org/10.7554/eLife.40986.009
**Source data 2.** Calculated mitochondrial accumulation data for *Figure 4a-b*
DOI: https://doi.org/10.7554/eLife.40986.010
**Source data 3.** Calculated mitochondrial accumulation data for *Figure 4c-d*
DOI: https://doi.org/10.7554/eLife.40986.011

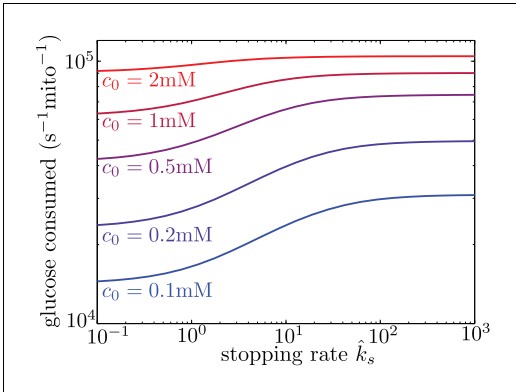

**Figure 5.** Mitochondrial stopping increases overall metabolic flux. Total glucose consumption per mitochondrion, averaged over the full interval, is shown for different edge glucose concentrations ($c_0$) as a function of the mitochondrial stopping rate $\hat{k}_s$. The limit of small $\hat{k}_s$ corresponds to uniform mitochondria distribution. Parameters for the model are taken from Table I. Source data is provided in '*Figure 5—source data 1*'.

DOI: https://doi.org/10.7554/eLife.40986.012

The following source data is available for figure 5:

**Source data 1.** Matlab code to calculate and plot total glucose consumption with localized glucose sources.

DOI: https://doi.org/10.7554/eLife.40986.013

However, neuronal cells are known to express a particularly efficient glucose transporter (GLUT3) (*Simpson et al., 2008*), and these transporters have been shown to be highly concentrated near the nodes of Ranvier (*Magnani et al., 1996*; *Rosenbluth, 2009*). We therefore assume that glucose import into the nodes is not rate limiting for myelinated neurons in physiological conditions. Introducing a finite rate of glucose transport would effectively decrease the intracellular glucose concentration at the nodes $c_0$, increasing the enhancement in metabolic flux due to mitochondrial localization. In subsequent sections, we explore the role of limited glucose import in unmyelinated axons with spatially uniform glucose permeability.

## Model for spatial organization in a glucose gradient

Extracellular brain glucose levels exhibit substantial regional variation, particularly under hypoglycemic conditions where more than ten-fold differences in local glucose concentrations have been reported (*Paschen et al., 1986*). Because individual neurons can traverse multiple different brain regions (*Matsuda et al., 2009*), a single axon can be subjected to heterogeneous glucose levels along its length. This raises the possibility that glucose-dependent mitochondrial localization can play a role in neuronal metabolic flexibility even in the case where glucose entry into the cell is not localized to distinct nodes. We thus extend our model to quantify the distribution of mitochondria in an axon with limited but spatially uniform glucose permeability that is subjected to a gradient of external glucose. This situation is relevant, for instance, to unmyelinated neurons in infant brains, as well as to in vitro experiments with neurons cultured in a glucose gradient (*Pekkurnaz et al., 2014*).

In this model, the extracellular environment provides a continuous source of glucose whose influx is limited by the permeability of the cell membrane. Intracellular glucose dynamics are then defined by the reaction-diffusion equation

$$\frac{dG}{dt} = D\frac{\partial^2 G}{\partial x^2} - k(x)G + P(x)(G_{\text{ext}}(x) - G) \tag{6}$$

where the first term corresponds to diffusive glucose spread, the second to a spatially varying metabolism of glucose, and the third to the entry of glucose into the cell. Here, $G_{\text{ext}}$ is the external glucose concentration, and $P(x)$ is the membrane permeability to glucose, which we assume to depend in a Michaelis-Menten fashion on the difference between external and internal glucose concentration:

$$P(x) = \frac{(2/r)PK_{MP}}{K_{MP} + |G_{\text{ext}}(x) - G(x)|} \tag{7}$$

where $P$ is the spatially uniform permeability constant in units of length per time. This functional form incorporates two known features of glucose transporters: (1) they are bidirectional, so that the overall flux through the transporter at low glucose levels should scale linearly with the difference between external and internal glucose (*Carruthers, 1990*); (2) neuronal glucose transporters saturate at high glucose levels (GLUT3 $K_{MP} \approx 3\text{mM}$ (*Maher et al., 1996*), with an even higher saturation constant for GLUT4 (*Nishimura et al., 1993*). When the difference in glucose levels is low, the overall

flux of glucose entering the cell reduces to $P(G_{\text{ext}}(x) - G(x))$. Mitochondria dynamics are defined as before (*Equation 4*), and we again assume Michaelis-Menten kinetics for glucose metabolism by hexokinase localized to mitochondria (*Equation 2*).

We note that the dynamics in *Equation 6* are governed by three time-scales: the rate of glucose transport down the length of the axon, rate of glucose consumption, and rate of glucose entry. The first of these rates becomes negligibly small in the limit $L \gg \sqrt{D(G + K_M)/(k_g \overline{M})}$. Because internal glucose levels can never exceed the external concentrations, in the range where $G_{\text{ext}}<10\text{mM}$, the rate of diffusive transport should become negligible for $L \gg 150\,\mu\text{m}$. In the limit where intracellular glucose is much less than $K_M$, this criterion reduces to $\hat{\lambda} \ll 1$, indicating that glucose diffuses over a very small fraction of the interval before being consumed. The interval length $L$ in this model represents an axonal length which can range over many orders of magnitude. We focus on axon lengths above several hundred microns, allowing us to neglect the diffusive transport of intracellular glucose (see Appendix 3).

The steady-state glucose profile can then be determined entirely by the local concentration of mitochondria and external glucose. For a given mitochondrial density $M(x)$ and external glucose profile $G_{\text{ext}}(x)$, the corresponding intracellular glucose concentration can be found directly by solving the quadratic steady-state version of *Equation 6* without the diffusive term. However, the steady-state mitochondrial distribution cannot be solved locally, because the limited number of mitochondria within the axon couples the mitochondrial density at different positions. We thus employ an iterative approach to numerically compute the steady-state solution for both glucose and mitochondrial density under a linear external glucose gradient $G_{\text{ext}} = G_{\text{min}} + (G_{\text{max}} - G_{\text{min}})\frac{x}{L}$ (see Materials and methods).

For parameter combinations where intracellular glucose concentrations are above $K_M$ but well below $G_{\text{ext}}$, the entry and consumption processes for glucose are both saturated. There is then a steep transition between two different regimes. In one regime, glucose entry exceeds consumption and internal glucose levels approach the external concentrations. In the other, consumption dominates and glucose levels drop below saturating concentrations. The key dimensionless parameter governing this transition can be defined as the ratio of entry to consumption rates:

$$\gamma = \frac{2P K_{MP} \overline{G}_{\text{ext}}}{k_g \overline{M} r (K_{MP} + \overline{G}_{\text{ext}})} \tag{8}$$

This ratio can be modulated in the cell either by recruiting varying amounts of glucose transporters (adjusting $P$) or changing the total amount of active hexokinase (adjusting $k_g \overline{M}$).

The remaining dimensionless parameters determining the behavior of this simplified model are the external glucose concentration relative to the hexokinase saturation constant ($\widehat{G}_{\text{ext}} = \overline{G}_{\text{ext}}/K_M$), the relative magnitude of the glucose gradient, $\Delta \widehat{G}_{\text{ext}} = (G_{\text{max}} - G_{\text{min}})/\overline{G}_{\text{ext}}$, the ratio of stopped to walking mitochondria $\hat{k}_s = k_s/k_w$, and the saturation constant for glucose transporters $K_{MP}/K_M \approx 96$. The last parameter is expected to remain approximately constant in neuronal cells. The average external glucose concentration and glucose gradient are expected to vary substantially depending on the glycemic environment to which the neuron is exposed. We note that $\Delta \widehat{G}_{\text{ext}}$ has a maximum possible value since the minimal glucose concentration cannot drop below 0zero. We proceed to analyze the limiting case where the glucose gradient is as steep as possible for any given value of average external glucose ($\Delta \widehat{G}_{\text{ext}} = 2$).

## Mitochondrial arrest enables metabolic enhancement under glucose gradient

We quantify the amount of mitochondrial accumulation at the high glucose side of the domain by calculating the total mitochondrial density within the distal 10% of the interval compared to a uniform distribution, in analogy to experimental measurements (*Pekkurnaz et al., 2014*). Substantial enrichment in the high glucose region occurs when glucose entry into the cell cannot keep up with consumption ($\gamma \ll 1$) and the intracellular glucose levels drop below the hexokinase saturation concentration $K_M$, as can be seen in the glucose and mitochondrial distributions plotted in *Figure 6a–c*. The interplay between external glucose levels and the entry/consumption rates is illustrated in

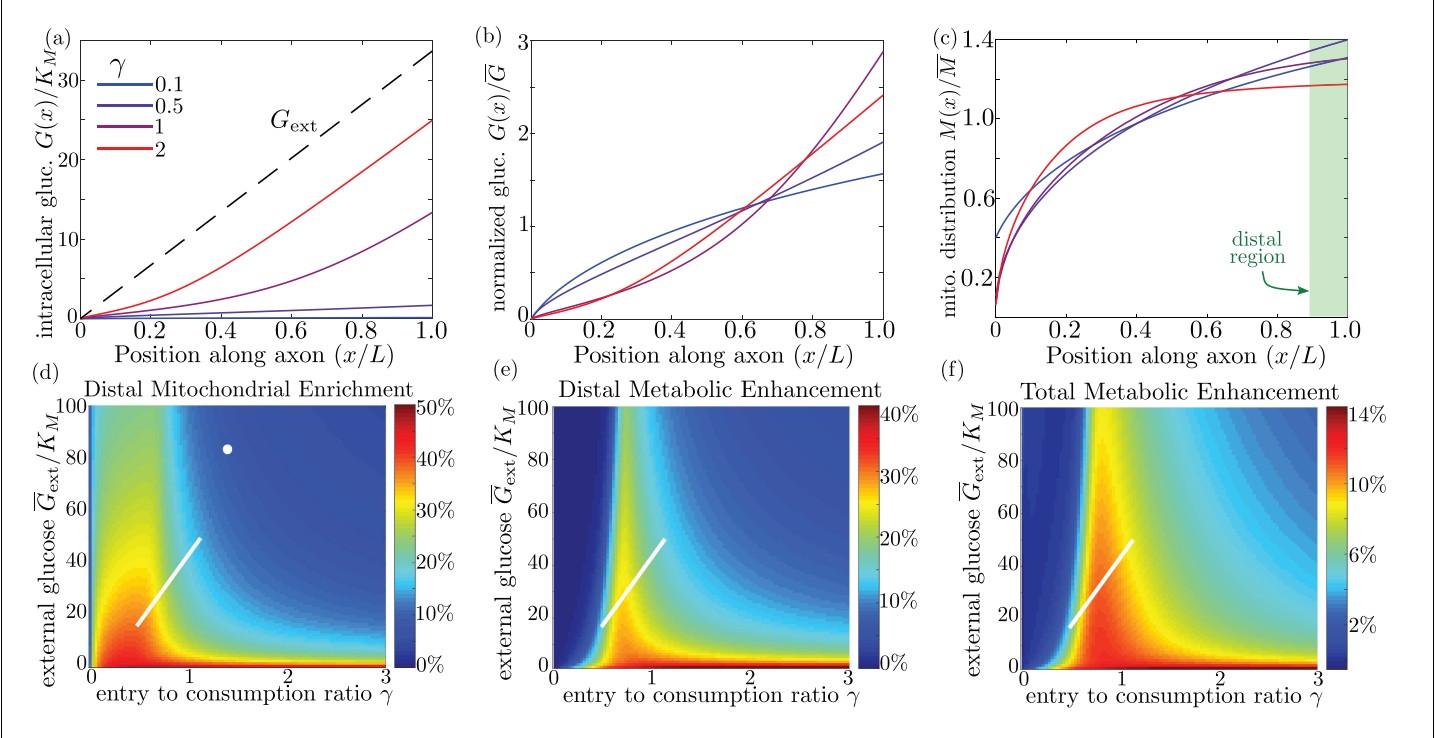

**Figure 6.** Mitochondrial and glucose organization in a region with uniform glucose permeability, subjected to a gradient of external glucose. (**a**) Internal glucose levels for the steady state solution with $\overline{G}_{ext}/K_M = 17$ ($\overline{G}_{ext} = 0.5$ mM) and varying ratios of entry to consumption rate $\gamma$. Black dashed line shows external glucose levels. (**b**) Corresponding normalized distribution of internal glucose. (**c**) Corresponding normalized mitochondrial distribution. Shaded box indicates distal region used for calculating mitochondrial enrichment and metabolic enhancement in panels (**d-e**). (**d**) Mitochondrial enrichment in the distal 10% of the interval at highest external glucose, compared to a uniform distribution. White dot marks estimated parameter values for neuronal cell culture experiments ($\overline{G}_{ext} = 2.5$ mM). (**e**) Enhancement in metabolic flux in the distal region at high glucose, compared to a uniform mitochondrial distribution. (**f**) Enhancement in metabolic flux over full interval. White line in (**d–f**) shows estimated parameter range for physiological glycemic levels $0.5\text{mM} < \overline{G}_{ext} < 1.5$ mM. Parameter values $\hat{k}_s = 19$, $\Delta\widehat{G}_{ext} = 2$ used throughout. Source data is provided in '*Figure 6—source datas 1–3*'.

DOI: https://doi.org/10.7554/eLife.40986.014

The following source data and figure supplement are available for figure 6:

**Source data 1.** Matlab code to calculate and plot results for model with linear glucose gradient.
DOI: https://doi.org/10.7554/eLife.40986.016
**Source data 2.** Calculated results for linear glucose gradient, with glucose-dependent mitochondrial halting.
DOI: https://doi.org/10.7554/eLife.40986.017
**Source data 3.** Calculated results for linear glucose gradient, with uniformly distributed mitochondria.
DOI: https://doi.org/10.7554/eLife.40986.018
**Figure supplement 1.** High stopping rate limit for model with uniform glucose permeability.
DOI: https://doi.org/10.7554/eLife.40986.015

*Figure 6d*. For external glucose concentrations well above $K_M$ there is a sharp transition to mitochondrial enrichment at $\gamma < 1$. At the lowest levels of intracellular glucose, accumulation is again reduced because a very small fraction of mitochondria are found in the stopped state. In the limit of high $k_s$, mitochondrial accumulation would occur for arbitrarily low values of $\gamma$ (*Figure 6—figure supplement 1*). We note that because glucose entry and turnover are much faster than diffusive spread for biologically relevant parameter regimes, the model results do not depend on the cell length $L$ (Appendix 3).

Experimental measurements of mitochondrial enrichment in cultured neurons subjected to a gradient of 0 to 5mM glucose have indicated an approximately 20% enrichment in mitchondrial counts at the axonal region exposed to high glucose. We note that using published estimates of typical glucose permeability and mitochondrial glucose turnover for mammalian cells (*Table 1*) yields a ratio of

entrance and consumption rates of $\gamma \approx 1.9$ for this experimental system. Because this ratio is above 1, we would not expect to see substantial mitochondrial enrichment. To result in the experimentally observed enrichment at high glucose, the ratio $\gamma$ would need to be reduced by approximately a factor of 2, implying the existence of additional regulatory mechanisms. Modulation of $\gamma$ could be achieved by either decreasing the number of glucose transporters in the cell (reducing $P$) or upregulating total hexokinase levels (increasing $k_g$). Neurons are believed to regulate both the density of glucose transporters and hexokinase activity in response to external glucose concentrations and varying metabolic demand (*Fujii and Beutler, 1985*; *Robey et al., 1999*; *Duelli and Kuschinsky, 2001*). In particular, adaptation to glycemic levels well above physiological values, as well as possibly reduced synaptic activity in a cultured environment, may result in downregulation of glucose transporters, lowering the value of $\gamma$. The discrepancy between model prediction and observed mitochondrial accumulation highlights the existence of additional regulatory pathways not included in the current model whose role could be explored in further studies that directly quantify glucose entry and consumption rates in cultured neurons.

Physiological brain glucose levels have been measured at 0.7 mM - 1.3 mM (*McNay et al., 2001*), with hypoglycemic levels dipping as low as 0.1 mM and hyperglycemic levels rising up to 4mM (*Silver and Erecińska, 1994*). Axons that stretch across different brain regions with varying glucose levels can thus be subject to a glucose gradient with $\overline{G}_{\text{ext}}$ on the order of 1 mM (white line on *Figure 6d*). We note that the physiological range overlaps substantially with the region of high mitochondrial accumulation, indicating that glucose-dependent halting can modulate mitochondrial distribution under physiologically relevant glycemic levels.

By accumulating mitochondria at the cellular region subjected to higher external glucose, the metabolic flux in that region can be substantially enhanced. In (*Figure 6e*) we plot the enhancement in glucose consumption rates (compared to the case with uniformly distributed mitochondria) within the $10\%$ of cellular length subjected to the highest glucose concentrations. Metabolic enhancement occurs within a narrow band of the $\gamma$ parameter. The drop-off in enhancement at low values of the internal glucose concentration (low $\gamma$) is due to the coupling between glucose levels and mitochondrial localization. Specifically, mitochondrial accumulation at the region subject to high glucose concentration increases the local rate of consumption in that region, driving down local internal glucose levels. Consequently, the difference in internal glucose concentrations between the two ends of the cell is decreased when internal levels fall substantially below $M$ (*Figure 6b*), reducing the enhancement of metabolic flux. Although mitochondrial accumulation decreases metabolic flux in the low glucose region, the total rate of glucose consumption integrated throughout the cell is enhanced by up to approximately 14% when $\gamma \approx 1$ (*Figure 6f*).

It is interesting to note that the typical physiological range of external glucose levels spans the narrow band of parameter space where metabolic enhancement is expected (white lines on *Figure 6e,f*). These results implicate glucose-dependent mitochondrial stopping as a quantitatively plausible mechanism of metabolic flexibility, increasing metabolism in regions with high nutrient availability for axonal projections that span between hypoglycemic and euglycemic regions. The magnitude of this effect can be tightly controlled by the cell through modulating overall rates of glucose entry and consumption. Thus, by coupling mitochondrial transport to local glucose levels, whole-cell changes in hexokinase or glucose transporter recruitment can be harnessed to tune the cell's response to spatially heterogeneous glucose concentrations.

## Discussion

The minimal model described here provides a quantitative framework to explore the interdependence of glucose levels and mitochondrial motility and their combined effect on neuronal metabolic flux. Glucose-mediated halting of mitochondrial transport is shown to be a plausible regulatory mechanism for enhancing metabolism in cases with spatially heterogeneous glucose availability in the neuron.

We have quantitatively delineated the regions in parameter space where such a mechanism can have a substantial effect on mitochondrial localization and metabolic flux. Specifically, mitochondrial positioning requires both sufficient spatial variation in intracellular glucose and sufficiently low absolute glucose levels compared to the saturation constant of the hexokinase enzyme. In the case of tightly localized glucose entry (as at the nodes of Ranvier), intracellular spatial heterogeneity requires

a small value of the dimensionless length scale for glucose decay ($\hat{\lambda} = \sqrt{DK_M/k_g\overline{M}L^2} \ll 1$). For physiologically estimated values, mitochondrial localization to the nodes is expected to occur for glucose levels below approximately 2 mM, comparable to physiological brain glucose concentrations (**McNay et al., 2001**; **John et al., 2008**). In the case where glucose can enter homogeneously throughout the cell surface (as with unmyelinated axons), heterogeneity can arise from an external glucose gradient. We show that metabolic enhancement through mitochondrial positioning occurs in a narrow range of the key parameter $\gamma = (2PK_{MP}\overline{G}_{\text{ext}})/(k_g\overline{M}(K_{MP} + \overline{G}_{\text{ext}}))$, which describes the ratio of glucose entry to glucose metabolism, and that this narrow range intersects with physiological estimates.

The model developed here is intentionally highly simplified, encompassing a minimal set of parameters necessary to describe glucose-dependent mitochondrial localization. Other regulatory pathways that determine mitochondrial positioning are not included in this basal model. In particular, we do not consider here calcium-based transport regulation, which is known to localize mitochondria to regions of synaptic activity (**Zhang et al., 2010**; **Wang and Schwarz, 2009**; **MacAskill and Kittler, 2010**; **Macaskill et al., 2009**). Upregulating OGT signaling in cultured cells has been shown to decrease the fraction of motile mitochondria by a factor of three, while reducing endogenous OGT nearly doubles the motile fraction, indicating that a substantial number of stationary mitochondria are stopped as a result of OGT activity (**Pekkurnaz et al., 2014**). Our model assumes the extreme case where all stopping events are triggered in a glucose-dependent manner, thereby isolating the effect of glucose heterogeneity. Stopping mechanisms dependent on neuronal firing activity could alter mitochondrial distribution in concert with glucose-dependent halting, increasing the density of mitochondria at presynaptic boutons or near areas of localized calcium influx as at the nodes of Ranvier (**Zhang et al., 2010**). We note that mitochondria have previously been shown to accumulate at spinal nodes of Ranvier in response to neuronal firing activity (**Fabricius et al., 1993**; **Zhang et al., 2010**). The mechanism described here provides an additional driving force for mitochondrial localization near the nodes even in quiescent neurons.

Additional metabolic feedback loops, not included in our model, may result in a more complex dependence of mitochondrial stopping on glucose concentration. In particular, both the pentose phosphate pathway and glycolysis generate intermediates that feed back into UDP-GlcNAc production by the hexosamine biosynthetic pathway (**Kruger and von Schaewen, 2003**; **Shirato et al., 2011**). Furthermore, several of the enzymes involved in the metabolic pathways linking glucose levels to Milton O-GlcNacylation may be regulated in a glucose-dependent manner. For example, the activity of the fructose-6-phosphate metabolizing enzyme GFAT is believed to be regulated by intermediates in the hexosamine pathway (**Traxinger and Marshall, 1991**) and O-GlcNAc transferase (OGT) itself is directly regulated by UDP-GlcNAc levels (**Hart et al., 2007**). Other enzymes, such as the de-GlcNAcylating enzyme OGA exhibit long term regulation of expression in response to altered glucose levels (**Zou et al., 2012**). These regulatory mechanisms provide additional potential routes of metabolic control through mitochondrial positioning.

Several key parameters that regulate mitochondrial localization in response to glucose heterogeneity can be dynamically regulated in neurons. Specifically, the rate of glucose consumption ($k_g\overline{M}$) can be tuned by modulating the concentration or activity of hexokinase within mitochondria or by altering total mitochondrial size and number. This parameter controls both the glucose decay length $\hat{\lambda}$ in the case of localized glucose influx and the ratio of glucose entry to consumption $\gamma$ in the case of spatially distributed entry. We note that our model assumes hexokinase to be localized exclusively to mitochondria. The predominant form of hexokinase in the brain (HK1) is known to bind reversibly to the mitochondrial membrane, with exchange between a mitochondria-bound and a cytoplasmic state believed to contribute to the regulation of its activity (**Golestani et al., 2007**). Release of hexokinase into the cytoplasm would result in more spatially uniform glucose consumption, negating the metabolic enhancement achieved through mitochondrial localization.

An additional parameter known to be under regulatory control is the rate of glucose entry into the neuron ($P$). The glucose transporters GLUT3 (**Simpson et al., 2008**; **Duelli and Kuschinsky, 2001**; **Weisová et al., 2009**) and GLUT4 (**Ashrafi et al., 2017**) have been shown to be recruited to the plasma membrane in response to neuronal firing activity. Interestingly, transporter densities are themselves spatially heterogeneous, concentrating near regions of synaptic activity (**Ashrafi et al., 2017**; **Ashrafi and Ryan, 2017**). The model described in this work quantifies the extent to which a

locally increased glucose influx can enhance total metabolic flux, given the ability of mitochondria to accumulate at regions of high intracellular glucose.

A number of possible feedback pathways linking glucose distribution and mitochondrial positioning are not included in our basic model. For instance, hexokinase release from mitochondria into the cytoplasm (potentially altering $k_g$) is known to be triggered at least in part by glucose-6-phosphate, the first byproduct in glucose metabolism (*Crane and Sols, 1954*). Chronic hypoglycemia has been linked to an upregulation in GLUT3 in rat neurons (*Uehara et al., 1997*), which would in turn lead to an increased glucose uptake ($P$). The fraction of glucose funneled into the hexosamine biosynthetic pathway (incorporated within $k_s$) can also be modified through feedback inhibition of GFAT by the downstream metabolic product UDP-GlcNAc (*Li et al., 2007*). Such feedback loops imply that several of our model parameters ($P$, $k_g$, $k_s$) are themselves glucose-dependent and may become spatially non-uniform in response to heterogeneous glucose. Incorporating these effects into a spatially resolved model of metabolism would require quantifying the dynamics of both the feedback pathways and mitochondrial positioning, and forms a promising avenue for future study.

Control of glucose entry and consumption underlies cellular metabolic flexiblity, and defects in the associated regulatory pathways can have grave consequences for neuronal health. Misregulation of hexokinase has been highlighted as a contributor to several neurological disorders, ranging from depression (*Regenold et al., 2012*) to schizophrenia (*Shan et al., 2014*). Neuronal glucose transporter deficiency has been linked to autism spectrum disorders (*Zhao et al., 2010*) and Alzheimer's disease (*Liu et al., 2008*). Furthermore, defects in mitochondrial transport, with the consequent depletion of mitochondria in distal axonal regions, contribute to peripheral neuropathy disorders (*Baloh, 2008*).

Glucose-dependent mitochondrial localization provides an additional layer of control, beyond conventionally studied regulatory mechanisms, which allows the cell to respond to spatial heterogeneity in glucose concentration. Our analysis paves the way for quantitative understanding of how flexible regulation of metabolism can be achieved by controlling the spatial distribution of glucose entry and consumption.

## Materials and methods

Source code (in MATLAB [*The MathWorks Inc, 2015*]) for all simulations and numerical calculations is available at: https://github.com/lenafabr/mitoManuscriptCodes (copy archived at https://github.com/elifesciences-publications/mitoManuscriptCodes).

### Discrete mitochondria simulations

We simulate the internodal space of the axon, between localized nodes of glucose entry, as a one-dimensional domain for a reaction diffusion system with motile reaction sinks. The glucose concentration field is discretized over 100 equidistant points along the domain. Its dynamics are governed by the reaction diffusion equation (*Equation 1*), evolved forward over time-steps of $\delta t$ using the forward Euler method. Because forward Euler methods have stringent conditions for stability and convergence, we use a time-step that is much smaller than both the glucose decay time-scale and the time-scale associated with diffusion over our spatially discretized grid (see below).

The number of mitochondria in the domain is calculated according to $N = \overline{M}L\pi r^2 \approx 38$, where the mitochondrial density $\overline{M}$, internodal distance $L$, and axonal radius $r$ are estimated from published data (*Table 1*; Appendix 1). The mitochondria are treated as discrete intervals of length $\Delta = 1$ μm, with the position of each mitochondrial center updated at each timestep. Over each time step, every motile mitochondrion moves a distance of $\pm v\delta t$, (with transport velocity $v = 1$ μm/s) and switches to a stationary state with probability $1 - \exp(-k_s\delta t)$, where $k_s(x)$ is a function of the center position of that mitochondrion (*Equation 3*). Mitochondria that reach within a distance of $\Delta/2$ from the ends of the domain are reflected, reversing their velocity while remaining motile. Analogously, every stationary mitochondrion switches to a motile state on each time-step with probability $1 - \exp(-k_w\delta t)$. Processive walks are initiated with equal probability in either direction.

At any given time, the spatial density of mitochondria is calculated from the location of mitochondrial centers at positions $x_1, \ldots x_N$, according to $M(x) = n(x)/(\pi r^2 \Delta)$, where

$$n(x) = \sum_{i=1}^{N} [\theta(x - x_i + \Delta/2) - \theta(x - x_i - \Delta/2)],$$

is the number of mitochondria overlapping spatial position $x$ and $\theta$ is the Heaviside step function.

We integrate the simulation forward in time-steps of $\delta t = 0.2 \frac{\Delta x^2}{D}$, where $\Delta x$ is the spatial discretization. This time-scale is much smaller than the relevant decay time for glucose consumption $\left[ \tau_g = \left( \frac{k_g \overline{M}}{K_M} \right)^{-1} \right]$. Using these small time-steps allows for stability and robust convergence with the forward Euler method. The simulation proceeds for $10^7$ steps. Simulations are repeated 100 times to obtain the histogram shown in *Figure 2*. Convergence to steady-state is established by comparing to calculations with the continuum model described in the subsequent sections.

## Mitochondrial distribution for spatially varying stopping rate

For an arbitrary spatial distribution of stopping rates $k_s(x)$ the corresponding steady-state mitochondrial distribution can be calculated directly by solving the equations for mitochondrial transport (*Equation 4*):

$$
\begin{aligned}
S &= \frac{k_s(x)(W_- + W_+)}{k_w} \\
v\frac{dW_+}{dx} &= \tfrac{1}{2}k_s(x)(W_- - W_+) \\
v\frac{dW_-}{dx} &= \tfrac{1}{2}k_s(x)(W_- - W_+).
\end{aligned}
\tag{9}
$$

Because our model assumes symmetry between anterograde and retrograde mitochondrial transport, as well as equal glucose concentrations at either boundary of the domain, we take $W_- = W_+$, implying that the population of walking mitochondria must be spatially constant. Consequently, the population of stopped mitochondria is proportional to the stopping rate ($S = Ck_s(x)/k_w$). The constant $C$ can be calculated from the normalization condition,

$$
\int_0^L M(x)dx = \int_0^L [W_-(x) + W_+(x) + S(x)]dx = \overline{M}L
\tag{10}
$$

The overall steady-state distribution of mitochondria is then given by,

$$
M(x) = W_-(x) + W_+(x) + S(x) = \frac{\overline{M}}{1 + \frac{1}{L}\int_0^L \frac{k_s(x)}{k_w}dx}\left[\frac{k_s(x)}{k_w} + 1\right]
\tag{11}
$$

Because the stopping rate is an explicit function of glucose concentrations $\left[k_s(x) = \frac{k_s G(x)}{K_M + G(x)}\right]$, this approach allows us to find the steady-state mitochondrial distribution for any fixed distribution of glucose.

## Numerical solution for steady-state distributions with localized glucose entry

We solve for steady-state glucose and mitochondrial distributions using a numerical method that evolves the glucose concentration forward in time while explicitly setting the mitochondrial concentration to its steady-state value at each step. The glucose distribution is initialized according to the steady-state solution for uniform consumption (*Equation 13*). Mitochondrial density $M(x)$ is calculated from the glucose distribution according to *Equation 11* and *Equation 3*.

The glucose distribution $G(x)$, in turn, evolves according to the mitochondrial distribution as given by *Equation 1* and *Equation 2*. The glucose profile is integrated forward with a timestep $\delta t = 10^{-5}L^2/D$. The distributions are assumed to be converged once the root mean squared rate of glucose change drops below the minimal cutoff: $10^{-6}k_g\overline{M}$. Results of the continuous mitochondrial distribution model are shown to match the discrete mitochondria simulations (*Figure 2b*). All subsequent analysis is done in the continuum limit.

## Analytical solution for low glucose limit

We validate our numerical calculations by comparing to the analytically tractable solution in the limit of low glucose and nearly uniform mitochondrial distribution. In the limit of spatially uniform, linear consumption, the steady-state reaction-diffusion equation for glucose can be expressed as

$$0 = D\frac{\partial^2 G}{\partial x^2} - kG(x) \tag{12}$$

where $k = k_g\overline{M}/K_M$ is the constant consumption rate.

Assuming fixed glucose concentrations ($c_0$) at the boundaries of the domain, the steady-state glucose distribution is then given by

$$G(x) = \frac{c_0\cosh(\frac{x}{\lambda})}{\cosh(\frac{L}{2\lambda})} \tag{13}$$

with $\lambda = \sqrt{\frac{D}{k}}$ defining the glucose decay length-scale. This quantity is a measure of how far glucose diffusively penetrates into the domain before being consumed by hexokinase. It is scaled by the size of the domain to give the dimensionless decay length scale $\hat{\lambda} = \sqrt{\frac{DK_M}{k_g\overline{M}L^2}}$ used as a key parameter in our model with localized glucose entry:

## Steady-state distribution with uniform permeability in the slow diffusion limit

For the model with spatially uniform glucose permeability, we solve directly for the steady state distributions of glucose and mitochondria in the limit of slow diffusivity. When diffusion along the domain is slow compared to the timescales of glucose consumption and glucose import, the steady-state equation for glucose concentration is given by a simplified form of *Equation 6*:

$$-k(x)G(x) + P(x)(G_{ext}(x) - G(x)) = 0 \tag{14}$$

Substituting $k(x) = \frac{k_g M(x)G(x)}{G(x)+K_M}$ and $P(x) = \frac{(2/r)PK_{MP}}{K_{MP}+|G_{ext}(x)-G(x)|}$, we get a quadratic equation in $G(x)$;

$$\left[1 - \frac{2PK_{MP}}{rk_gM}\right]G(x)^2 + \left[\frac{2PK_{MP}G_{ext}}{rk_gM} - \frac{2PK_{MP}K_M}{rk_gM} - G_{ext} - K_{MP}\right]G(x) + \left[\frac{2PK_{MP}K_MG_{ext}}{rk_gM}\right] = 0 \tag{15}$$

For a given mitochondrial profile, this quadratic equation is solved to find $G(x) = G(M(x))$. The mitochondrial distribution, $M(x)$ is then updated according to *Equation 11* and *Equation 3*. We thus arrive at an iterative solution for $G(x)$ and $M(x)$.

## Acknowledgements

We thank Saurabh Mogre for fruitful discussions, Manho Tang for numerical modeling advice, and David Kleinfeld for helpful comments on the manuscript.

## Additional information

### Funding

| Funder | Grant reference number | Author |
| --- | --- | --- |
| National Institutes of Health | R35GM128823 | Gulcin Pekkurnaz |
| University of California, San Diego | Chancellor's Research Excellence Scholarship | Anamika Agrawal |
| Alfred P. Sloan Foundation | FG-2018-10394 | Elena F Koslover |

The funders had no role in study design, data collection and interpretation, or the decision to submit the work for publication.

## Author contributions

Anamika Agrawal, Conceptualization, Software, Formal analysis, Validation, Investigation, Visualization, Methodology, Writing—original draft, Writing—review and editing; Gulcin Pekkurnaz, Conceptualization, Supervision, Funding acquisition, Methodology, Writing—review and editing; Elena F Koslover, Conceptualization, Software, Formal analysis, Supervision, Funding acquisition, Validation, Investigation, Visualization, Methodology, Writing—original draft, Project administration, Writing—review and editing

## Author ORCIDs

Anamika Agrawal http://orcid.org/0000-0002-1213-2321
Elena F Koslover http://orcid.org/0000-0003-4139-9209

## Decision letter and Author response

Decision letter https://doi.org/10.7554/eLife.40986.031
Author response https://doi.org/10.7554/eLife.40986.032

# Additional files

## Supplementary files

• Transparent reporting form
DOI: https://doi.org/10.7554/eLife.40986.019

## Data availability

MATLAB code for implementing the models described in this study has been made available on Github: https://github.com/lenafabr/mitoManuscriptCodes (copy archived at https://github.com/elifesciences-publications/mitoManuscriptCodes). Source data files for Figures 3, 4, 5, 6 and the appendix figure are provided in the manuscript and supporting files.

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

## Appendix 1

DOI: https://doi.org/10.7554/eLife.40986.020

## Estimating physiological parameter values

In this appendix we describe our approach to estimating the parameter values summarized in *Table 1* from published experimental data.

### Glucose diffusivity ($D$)

Glucose is a small molecule of comparable molecular weight to ATP, which has a diffusion coefficient of 140 µm²/s (*Mironov, 2007*; *Vendelin et al., 2000*)

### Glucose consumption rate per mitochondrion ($k_g$)

The oxidative capacity of muscle mitochondria has been measured at 5.8 mL of $O_2$ per min per mL mitochondria (*Harris and Attwell, 2012*; *Schwerzmann et al., 1989*). We assume 6 glucose molecules are consumed per molecule of oxygen, and a volume of 0.3 µm³ for globular mitochondria (*Posakony et al., 1977*). The corresponding glucose turnover rate of a mitochondrion is then calculated as $1.3 \times 10^5$ glucose per second per mitochondrion.

### Axon radius ($r$)

The thickness of mammalian brain axons varies widely from 0.1 µm to 10 µm (*Perge et al., 2012*). Statistical measurements in the human brain show that most axon diameters fall below 1 µm, with a long-tailed distribution of substantially thicker axons (*Liewald et al., 2014*). We take as our estimate a median diameter of 0.8 µm, which is consistent with measurements in human brain regions (*Liewald et al., 2014*), in the rat corpus collosum (*Barazany et al., 2009*), guinea pig retinal neurons (*Perge et al., 2009*), and in a number of other mammalian tracts (*Perge et al., 2012*).

### Internodal distance ($L$)

Typical internodal lengths vary widely from 200 µm to 1500 µm (*Jacobs, 1988*; *Court et al., 2004*). We use a value of $L = 250$ µm as measured in the axons of rat anterior medullary velum (*Ibrahim et al., 1995*).

### Mitochondrial density ($\overline{M}$)

Measurements of mitochondrial concentration in human spinal muscular nerves give a linear density of about 15 mitochondria per 100 µm of axon (*Xu et al., 2016*). Similar densities are observed in *Figures 1* and *2* of (*Pekkurnaz et al., 2014*). Assuming an axonal radius of $r \approx 0.4$ µm gives a corresponding density of 0.3 µm⁻³. EM measurements in rat brain neurons indicate that mitochondria occupy approximately $8\%$ of the neuronal cytoplasmic volume (*Pysh and Khan, 1972*). Assuming a mitochondrial volume of 0.3 µm⁻³ (*Posakony et al., 1977*) would give the same density estimate of 0.3 mitochondria per µm³.

### Hexokinase Michaelis-Menten constant ($K_M$)

The Michaelis-Menten constant for glucose phosphorylation by the neuronal isoform of hexokinase (HKI) has been measured as $K_M = 0.03$mM (*Wilson, 2003*).

### Ratio of stopped to moving mitochondria ($k_s/k_w$)

In *Pekkurnaz et al., 2014*, mammalian neurons grown under high (30 mM) glucose conditions were found to have mitochondria that spent approximately 5% of their time in motion. This fraction should correspond to $k_w/(k_s + k_w) \approx 0.05$ under our simplified model for mitochondrial motility.

### Membrane permeability to glucose ($P, K_{MP}$)

The neuronal glucose transporter GLUT3 in rat cerebellar granule neurons has been measured to have a turnover rate of $k_{\mathrm{glut3}} = 853\,\mathrm{s}^{-1}$ and a Michaelis-Menten constant of $K_{M,\mathrm{glut3}} = 3\,\mathrm{mM}$ (*Maher et al., 1996*). In the same study, the density of GLUT3 channels was measured as 18 pmol/mg cell membrane. We assume the cell membrane has a density of order 1 g/cm$^3$ and forms a sheet of thickness 4 nm. This allows us to calculate the area density of GLUT3 channels in cerebellar neurons as approximately $a$ = 43 transporters/µm$^2$.

In the case where the difference in external and internal glucose concentration ($\Delta G$) is below $K_{M,\mathrm{glut3}}$, we can approximate the net flux into the cell as,

$$\mathrm{flux} = k_{\mathrm{glut3}} a \frac{\Delta G}{K_{M,\mathrm{glut3}}} = P\Delta G, \tag{16}$$

allowing an estimate of the permeability $P = \frac{k_{\mathrm{glut3}} a}{K_{M,\mathrm{glut3}}} \approx 0.02\,\mu\mathrm{m/s}$

## Appendix 2

DOI: https://doi.org/10.7554/eLife.40986.020

### Effective Michaelis-Menten kinetics for glycosylation of Milton

We assume individual steps in glucose metabolism follow classic Michaelis-Menten kinetics, with the rate of product formation given by $dP/dt = v_i S/(K_{Mi} + S)$. We further assume that all pathways considered here are operating in steady-state, with a stationary concentration of all intermediates. When several Michaelis-Menten reactions are connected in series (*eg: A →B →C*), steady state requires that the dependence of final product formation $C$ on the initial reactant $A$ is given by,

$$\frac{dC}{dt} = \frac{v_{AB}A}{K_{AB} + A} \tag{17}$$

where $v_{AB}, K_{AB}$ are the maximum rate and saturation constant for the initial $A →B$ reaction.

If two pathways branch from a single intermediate, as occurs when the hexosamine biosynthetic pathway splits off first from the pentose phosphate pathway and then from glycolysis, then we have an additional reaction $B →D$ that alters the rate of $C$ formation. We make the key assumption that the saturation constants in the first step of both branching pathways are comparable ($K_{BC} \approx K_{BD}$). Steady state then requires

$$\frac{v_{AB}A}{K_{AB} + A} = \frac{(v_{BC} + v_{BD})B}{K_{BC} + B}$$
$$\frac{dC}{dt} = \left(\frac{v_{BC}}{v_{BC} + v_{BD}}\right)\frac{v_{AB}A}{K_{AB} + A} \tag{18}$$

Thus, if the saturation concentrations of the splitting reactions are similar, then the formation of the final product occurs at a rate proportional to the initial substrate consumption, with the same saturation constant. As illustrated in the pathway schematic (*Appendix 2—figure 1B*), the branching of both the pentose phosphate pathway and glycolysis from the pathway leading to UDP-GlcNAc formation involves similar values of the Michaelis-Menten constant. We therefore assume that the rate of Milton glycosylation by OGT ($dC/dt$) is proportional to the rate of initial glucose consumption by hexokinase ($|dA/dt|$). This assumption justifies our use of the same $K_M$ for both glucose consumption and mitochondrial stopping. The fraction of metabolic flux funneled into Milton glycosylation is subsumed into the effective rate constant $k_s$.

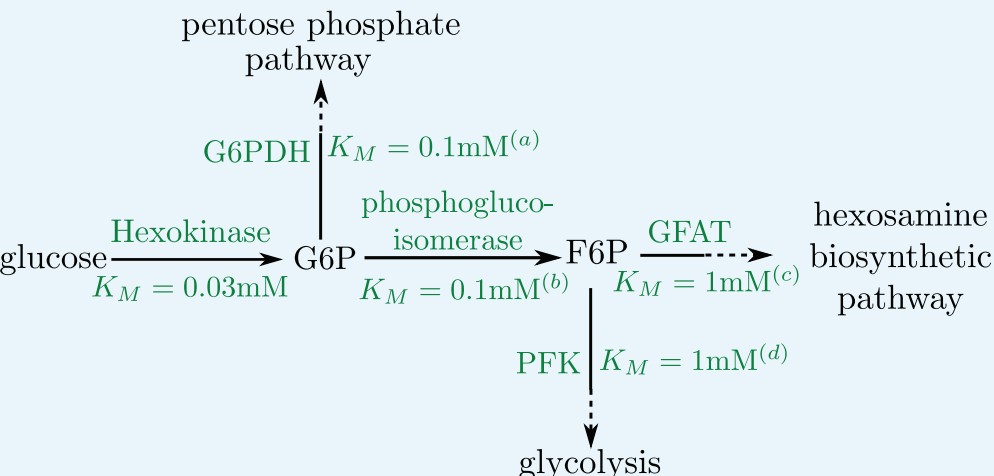

**Appendix 2—figure 1.** Schematic of early pathway branches in glucose metabolism, showing

the branching of the pentose phosphate pathway and glycolysis from the hexosamine biosynthetic pathway that leads to UDP-GlcNAc formation. Saturation concentrations are labeled for each of the initial branching reactions. Note that in both cases, the splitting branches have comparable values of $K_M$ (a) *Duffieux, 2000*; (b) *Kahana et al., 1960*; (c) *Li et al., 2007*; (d) *Urbina and Crespo, 1984*.

DOI: https://doi.org/10.7554/eLife.40986.022

## Appendix 3

DOI: https://doi.org/10.7554/eLife.40986.020

### Effect of domain length $L$ in uniform permeability model

Dimensional analysis of **Equation 6** indicates that the diffusive term for glucose dynamics will be negligible compared to the consumption and entry terms in the limit $L \gg \sqrt{D(G + K_M)/(k_g \overline{M})}$. We assume that external glucose concentrations are well below 10mM, indicating that the diffusive term is irrelevant for $L \gg 150\,\mu\text{m}$. If diffusion is neglected, the only length units in the model are found within external glucose and mitochondrial concentrations, both of which are fixed parameters independent of axonal length. We therefore expect in this limit that the model results will not depend on the interval length $L$.

To verify the accuracy of this limit, we plot glucose and mitochondrial distributions for the full model (including diffusion) for interval lengths of $L = 100\,\mu\text{m}$, 1 mm, and 1 cm (**Appendix 3—figure 1a,b**). All other parameters are from our physiological estimates in **Table 1**. We note that for lengths well above $100\mu\text{m}$, the distributions are independent of $L$ and are nearly identical to those expected for the model with diffusion excluded. We also plot mitochondrial accumulation and metabolic enhancement in the distal 10% of the interval obtained from solutions of the full model with diffusive transport, which match well to the plots in the main text (**Figure 6**) that neglect diffusive transport. We can thus assume that glucose entry and turnover are much faster than diffusive spread for biologically relevant parameter regimes.

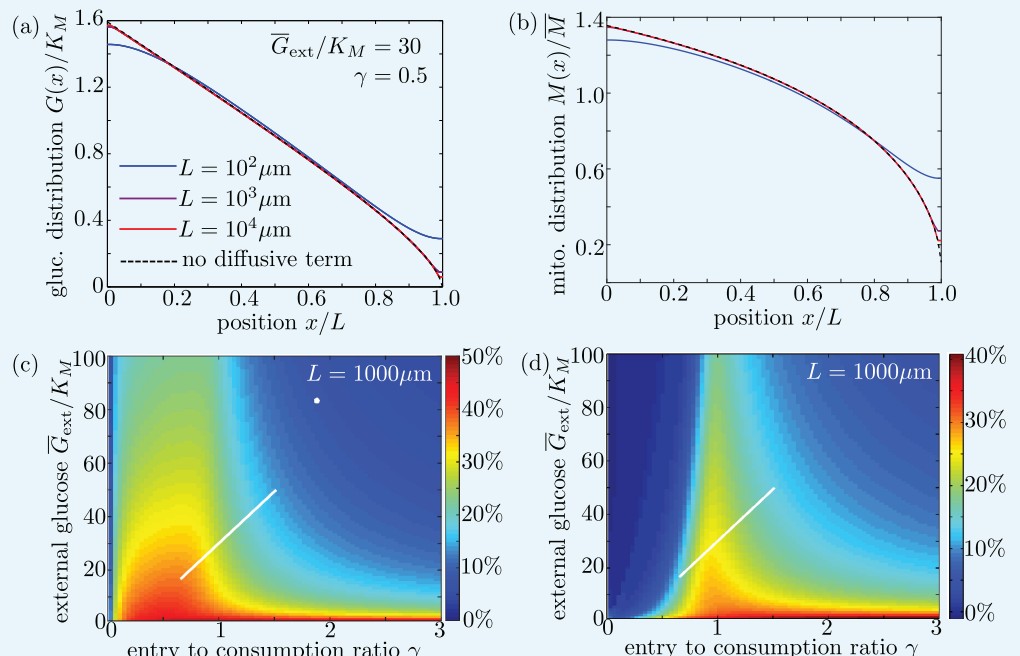

**Appendix 3—figure 1.** Long cell length limit can be approximated by a simplified model without diffusive transport. (**a-b**) Steady-state intracellular glucose and mitochondrial distributions from numerical solutions of **Equation 6** using different values of interval length $L$. Black dashed line shows solution of the simplified model with the diffusive term removed. (**c**) Mitochondrial enrichment in the distal 10% of the interval subject to highest external glucose, as compared to a uniform distribution. White line indicates physiological brain glucose levels, while white dot indicates glucose levels in cultured neurons (**Pekkurnaz et al., 2014**). (**d**) Enhancement in metabolic flux in the distal 10% of the interval, compared to uniform mitochondrial distribution. Plots (**c**) and (**d**) are obtained from solutions of the full model with

diffusive transport and are indistinguishable from (*Figure 6d–e*) for an interval length of 1000 μm.

DOI: https://doi.org/10.7554/eLife.40986.024

The following source data is available for figure :

**Appendix 3—figure 1—source data 1.** Matlab code to calculate and plot results for finite length domains with limited glucose permeability.
DOI: https://doi.org/10.7554/eLife.40986.025

**Appendix 3—figure 1—source data 2.** Calculated data for domain length 100 $\mu$m.
DOI: https://doi.org/10.7554/eLife.40986.026

**Appendix 3—figure 1—source data 3.** Calculated data for domain length 1000 $\mu$m.
DOI: https://doi.org/10.7554/eLife.40986.027

**Appendix 3—figure 1—source data 4.** Calculated data for domain length 1000 $\mu$m with uniformly distributed mitochondria.
DOI: https://doi.org/10.7554/eLife.40986.028

**Appendix 3—figure 1—source data 5.** Calculated data for domain length 10000 $\mu$m.
DOI: https://doi.org/10.7554/eLife.40986.029

