## [Decision Letter]

Thank you for submitting your article "Spatial control of neuronal metabolism through glucose-mediated mitochondrial transport regulation" for consideration by *eLife*. Your article has been reviewed by two peer reviewers, one of whom is a member of our Board of Reviewing Editors, and the evaluation has been overseen by Naama Barkai as the Senior Editor. The following individual involved in review of your submission has agreed to reveal his identity: Timothy A Ryan (Reviewer #1).

The reviewers have discussed the reviews with one another and the Reviewing Editor has drafted this decision to help you prepare a revised submission.

Summary:

This manuscript describes a general framework for examining the quantitative details of how the feed-forward control of mitochondrial motility by a downstream sensing of glucose metabolism (through OGlcNAc modification) might lead to both changes in the spatial distribution of mitochondria and the potential enhancement of overall metabolic flux that would ensue. The referees agreed that the manuscript was interesting, well-written, and timely.

Essential revisions:

1) The treatments seem to treat the OGT modification as irreversible as no terms describing the reverse reaction that would be driven by an O0GlcNAcase.

2) The authors also only consider two possible fates of Glucose 6P (glycolysis and the hexosamine pathway). The pentose phosphate pathway is another possible shunt that should probably at the very least be discussed or its omission justified.

3) A forward Euler scheme was used to solve the reaction-diffusion Equation 1. Such schemes are known to be generally unstable, so the authors need to justify their use and/or re-do the numerics with a proper scheme (e.g. Crank-Nicholson) to confirm the results.

---

## [Author Response]

Essential revisions:1) The treatments seem to treat the OGT modification as irreversible as no terms describing the reverse reaction that would be driven by an O0GlcNAcase.

We thank the reviewer for this comment and agree that this was not clearly explained in the manuscript. In our simplified quantitative model, we assume that all pauses are associated with an O-GlcNAcylation event. Recovery from the pause at the constant rate k_w_ represents the removal of the modification of Milton through the activity of the complementary enzyme O-GlcNAcase. We have added a paragraph to our manuscript more thoroughly explaining the rationale behind our simplified model for mitochondrial dynamics, quoted below:

“It has recently been demonstrated that the key motor adaptor protein (Milton) is sensitive to glucose levels, halting mitochondrial motility when it is modified through OGlcNAcylation by the OGT enzyme [Pekkurnaz et al., 2014]. […] Other sources of pausing, such as Ca^2+^-regulated motor disengagement, PINK1/Parkin-mediated detachment of motors, and anchoring to the microtubules by syntaphilin [Schwarz, 2013], are not considered here in order to focus specifically on the effect of glucose-dependent mitochondrial spatial organization.”

Furthermore, we include an additional paragraph in the Discussion pointing out simplifications in this model that ignore potential glucose-dependent regulation of other steps in the metabolic pathway connecting glucose and mitochondrial stopping:

“Additional metabolic feedback loops, not included in our model, may result in a more complex dependence of mitochondrial stopping on glucose concentration. […] These regulatory mechanisms provide additional potential routes of metabolic control through mitochondrial positioning.”

2) The authors also only consider two possible fates of Glucose 6P (glycolysis and the hexosamine pathway). The pentose phosphate pathway is another possible shunt that should probably at the very least be discussed or its omission justified.

We thank the reviewer for pointing out this omission. We have now included the metabolic branch for the pentose phosphate pathway in our schematic in Figure 1C. We note that pathway branches where the saturation constants are similar to the corresponding step in the HBP, will not affect the overall kinetics in our model, except by changing the fraction of glucose that flows down the HBP pathway. This fraction is subsumed in the effective pausing rate k_s_. We have modified the relevant paragraph in the model development section to address this issue:

“Upon entry into the cell, the first rate-limiting step of glucose metabolism is its conversion into glucose-6-phosphate by hexokinase. […] We therefore model the kinetics of Milton modification using the same Michaelis-Menten form as for hexokinase activity, with the pathway flux leading to Milton modification subsumed within a rate constant for mitochondrial stopping (k_s_).”

We also include a detailed justification of using the same saturation constant K_M_ for both glucose consumption and O-GlcNAcylation of Milton in Appendix 2. A schematic with appropriate citations has been added to Appendix 2, listing the saturation constants of both the pentose phosphate and glycolysis branching pathways.

3) A forward Euler scheme was used to solve the reaction-diffusion Equation 1. Such schemes are known to be generally unstable, so the authors need to justify their use and/or re-do the numerics with a proper scheme (e.g. Crank-Nicholson) to confirm the results.

We thank the reviewer for the comments, and agree that forward Euler methods have poor convergence properties. However, such methods can be used for simple systems, provided a sufficiently small time-step is used to assure convergence (smaller than the fastest time-scale in the system, including the scale set by the spatial discretization). We know convergence is achieved in our model because the magnitude of the time derivative goes to zero as the simulation approaches steady state. To clarify this issue, we have added additional description of our choice of timestep and convergence criteria in the Materials and methods section, under “Discrete mitochondria simulations”:

“Because forward Euler methods have stringent conditions for stability and convergence, we use a time-step that is much smaller than both the glucose decay time-scale and the time-scale associated with diffusion over our spatially discretized grid (see below).”

“We integrate the simulation forward in time-steps of δt=0.2∆x2D, where ∆x is the spatial discretization. This time-scale is much smaller than the relevant decay time for glucose consumption τg=kgM-KM-1. Using these small time-steps allows for stability and robust convergence with the forward Euler method.”

In the section titled “Numerical solution for steady-state distributions with localized glucose entry” we note the convergence criterion for the continuum model: “The distributions are assumed to be converged once the root mean squared rate of glucose change drops below the minimal cutoff: 10^−6^*k_g_M*.”